# A plausible mechanism for longitudinal lock-in of the plant cortical microtubule array after light-induced reorientation

## Original Research Article

Microtubule dynamics; Cortical microtubule array; Katanin; Cytoskel et al self-organization; Stochastic modelling; Theory and Computation.

## Author for correspondence:
M. Saltini,
E-mail: marco.saltini@ebc.uu.se

Marco Saltini[1]  and Bela M. Mulder[2]

[1] Department of Ecology and Genetics, Animal Ecology, Uppsala University, Uppsala, Sweden; [2] Living Matter Department, AMOLF, Amsterdam, The Netherlands

## Abstract

The light-induced reorientation of the cortical microtubule array in dark-grown *Arabidopsis thaliana* hypocotyl cells is a striking example of the dynamical plasticity of the microtubule cytoskeleton. A consensus model, based on *katanin*-mediated severing at microtubule crossovers, has been developed that successfully describes the onset of the observed switch between a transverse and longitudinal array orientation. However, we currently lack an understanding of why the newly populated longitudinal array direction remains stable for longer times and re-equilibration effects would tend to drive the system back to a mixed orientation state. Using both simulations and analytical calculations, we show that the assumption of a small orientation-dependent shift in microtubule dynamics is sufficient to explain the long-term lock-in of the longitudinal array orientation. Furthermore, we show that the natural alternative hypothesis that there is a selective advantage in severing longitudinal microtubules, is neither necessary nor sufficient to achieve cortical array reorientation, but is able to accelerate this process significantly.

## 1. Introduction

The cortical microtubule array (hereafter CA) is a highly ordered structure occurring in cells of higher plants, which plays a key role in the growth-driven morphogenesis of cells and, therefore, also of the organism as a whole. It is known that the function of the CA is intimately linked to its spatial organisation which, in turn, is linked to its dynamic properties (Allard et al., 2010; Dixit & Cyr, 2004; Eren et al., 2010; Tindemans et al., 2010). In contrast to animal cells, the CA is built and reorganised without benefit of a microtubule organising centre (centrosome). Instead, new microtubules in the CA are generated in two distinct ways: from nucleation templates—that is, γ-tubulin complexes, mostly located on the lattice of already existing microtubules (Chan et al., 2009; Murata et al., 2005), or from severing events of already-existing microtubules. The latter mode of generation of new microtubules is mediated by the severing protein *katanin* that localises at microtubule crossovers and preferentially severs the newer one, that is, the overlying one (Lindeboom et al., 2013). It plays a crucial role in the reorientation of the cortical array as a response to blue light. Typically, the direction of the cortical microtubule array of growing plant cells is transverse to the long axis of the cell. However, upon exposure to blue light, in dark-grown hypocotyl cells of *A. thaliana*, the initially transverse array undergoes a striking reorientation to a direction longitudinal to the long axis of the cell. Since the initial array is transverse to the long axis of the cell, newer microtubules at crossovers are most likely longitudinal and, hence, the occurrence of multiple severing events quickly generates a new exponentially growing population of longitudinal microtubules (Lindeboom et al., 2013).

Experimental and computational studies have identified the crucial role of the stability of the dynamic microtubule ends in enabling CA reorientation (Lindeboom et al., 2019; Nakamura et al., 2018). Furthermore, a recent theoretical study has shown that the combination of preferential severing and a high probability of stabilisation-after-severing for the newly-created microtubule plus-ends is a necessary ingredient for the reorientation process to start (Saltini & Mulder, 2020). However, these studies focused on the first phase of the reorientation process,

when the initial transverse array can still be seen as a constant background and free tubulin to build new microtubules is likely available in abundance. While these assumptions are realistic at the start of the reorientation process, at a later phase the amount of free tubulin is bound to become scarce as the number of growing microtubules increases. As transverse microtubules are also dynamic, this scarcity of tubulin would imply their gradual depolymerisation, as they are outcompeted by the large population of growing longitudinal microtubules. This in turn, however, decreases the opportunity to create new crossovers and, therefore, to create new longitudinal microtubules from severing events. This suggests that while the initial asymmetry in the number of preferential severing of microtubules in the longitudinal direction is a sufficient ingredient to yield an initial asymmetry between the two populations of differently oriented microtubules, over time, this bias is expected to fade. New transverse microtubules will start to be nucleated, and their crossovers with preexisting longitudinal microtubules can now serve to generate more transverse ones, through the same severing mechanism. In fact, all things being equal, one would expect the system to evolve to a novel steady state with an equal number of transverse and longitudinal microtubules. This clearly contrasts with the experimental findings of an ultimately stable longitudinal array, and raises the question of the mechanism by which this switch in orientations can be maintained.

Here, we explore the hypothesis that the ultimate asymmetry between differently oriented microtubules could be a consequence of small differences in their dynamic behaviour. The idea that the dynamics of microtubules can be influenced by cell geometry has received a lot of attention since the seminal work of Hamant et al. (2008) (for a review see Landrein & Hamant, 2013) which provided the first evidence that microtubule would in fact prefer to align in the direction of maximal stress in the cell wall. In the common cylindrical plant cell geometry, this direction of maximal stress follows the direction of strongest curvature, which thus provides a possible explanation for the ubiquitous transversality of the CA in this cell geometry. Although the severing protein katanin is implicated in mediating the interaction between wall stress and microtubule stabilisation (Uyttewaal et al., 2012), the precise mechanism by which this occurred has not been elucidated to date. Computational models of cortical microtubule dynamics on closed surfaces with different geometries have also shown that slight directional cues, caused, for example, by catastrophes induced at sharp cell edges, relative stabilisation on specific cell faces or increased number of microtubules generated in one specific direction, are sufficient to select a single preferential direction of ordering of the CA as a whole (Ambrose et al., 2011; Chakrabortty et al., 2018; Mirabet et al., 2018; Sambade et al., 2012). On the basis of these considerations, we posit that a mechanism by which microtubule dynamics is sensitive to orientation with respect to a cell axis is possible, and that such a bias can impact the global organisation of the CA. Specifically, we will assume that microtubules in the longitudinal direction will grow slightly faster than ones in the transverse direction, but, as we show, an analogous decrease of catastrophe rate in the longitudinal direction will serve the same purpose.

We implement our hypothesis in a stylised stochastic model of dynamic microtubules that can only occur in two distinct orientations—that is, transverse and longitudinal. These two populations of microtubules compete for the same pool of available tubulin dimers as their building material. Besides the indirect interaction through the available tubulin pool, the two populations of microtubules directly interact through severing events at crossovers. Both simulations and additional analytical calculations show that the small difference in the dynamic parameters between the two populations can explain the experimentally observed lock-in of the CA to the longitudinal direction after reorientation from an initially transverse state. We also test an alternative hypothesis that the preferential severing of longitudinal microtubules is specifically required for the reorientation to occur. While we show that the latter mechanism is neither a necessary nor a sufficient ingredient for the reorientation to occur, we do find that it can significantly increase the speed of the reorientation.

## 2. Methods

To test our hypothesis that the full and subsequently maintained reorientation of the CA (see Figure 1a) is caused by a small asymmetry in the dynamics of differently oriented microtubules, we introduce a stochastic model for microtubules undergoing dynamic instability. We focus on the generation of new microtubules through nucleation and severing. A full mathematical description of the system of differential equations controlling the dynamics of individual microtubules and the steady-state solution for microtubule length distribution can be found in the Supplementary Information S1.

### 2.1. The model

The model, based on the Dogterom–Leibler model for microtubule dynamics (Dogterom & Leibler, 1993) consists of two populations of microtubules undergoing dynamic instability, the longitudinal ($M_\parallel$) and transverse population ($M_\perp$), respectively. The two populations of microtubules do not have a specific localisation in space, that is, they exist in two abstract microtubule spaces in which the actual geometry of the cell does not play a role. As a matter of fact, $\parallel$ and $\perp$ are purely labels in our model. Nevertheless, for the sake of clarity and given the system our model is applied to, we will refer to the two populations of microtubules as longitudinal and transverse.

**2.2. Tubulin redistribution.** The two populations compete for a finite tubulin pool, that is, they can only access a finite number of tubulin-dimers to fuel their growth or de novo nucleation. Let $L_{\text{tot}}$ be the total amount of tubulin in the system, expressed as the maximum total length of microtubules to which it could give rise. Then, $L_{\text{tot}}$ is divided over three different populations: the free tubulin pool $L_f$, the tubulin incorporated into longitudinal microtubules $L_\parallel$, and the tubulin incorporated into transverse microtubules $L_\perp$, such that

$$L_{\text{tot}} = L_f + L_\parallel + L_\perp, \tag{1}$$

see Figure 1b for the finiteness of the tubulin pool has immediate consequences for the dynamics of microtubules. In particular, the abundance of free tubulin is positively correlated to both the nucleation rate and the growing speed of microtubules.

**2.3. Nucleation of new microtubules.** Experiments (Wieczorek et al., 2015) have identified a Hill-type dose–response relation between the nucleation rate of new microtubules and the abundance of free tubulin, that is,

$$R_n\left(L_f\right) = r_n \frac{L_f^a}{L_f^a + L_v^a}, \tag{2}$$

where $a \simeq 6$, $L_v$ is a constant of dimension length that governs the cross-over from a diffusion limited regime at low free tubulin densities to an intrinsic association rate limited regime at high free

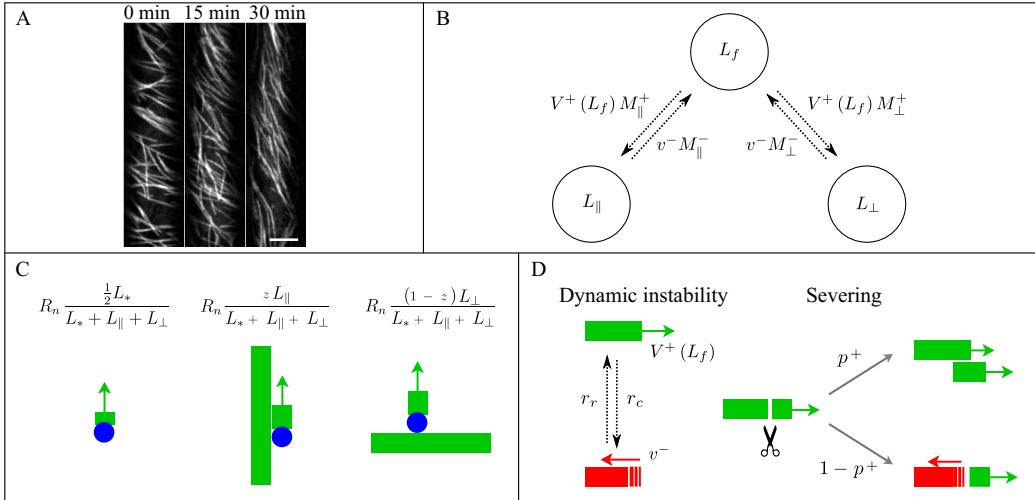

**Fig. 1.** (a) Microtubule reorientation of dark-grown hypocotyl cells at 0, 15, and 30 min after induction of reorientation by blue light. Scale bar, 5 μm. (b) Overall dynamics of the two microtubule populations and tubulin redistribution in the system. Free tubulin is recruited by microtubules of the two dynamic populations with a speed proportional to the total number of growing microtubules, while it returns to the free pool with a speed proportional to the total number of shrinking microtubules. The label ± stands for growing/shrinking, respectively. (c) Nucleation rates for new longitudinal microtubules. From left to right, new microtubules can be nucleated through dispersed nucleation, microtubule-based nucleation parallel to the mother longitudinal microtubule, or microtubule-based nucleation orthogonal to the mother transverse microtubule. Blue circles represent the $\gamma$-tubulin complex. (d) Dynamics of an individual microtubule. Microtubules undergo dynamic instability and are severed with rate proportional to their length. Newly-created plus end after severing enters either the growing state with probability $p^+$, or the shrinking state with probability $1 - p^+$.

tubulin densities, and $r_n$ is the nucleation rate of new microtubules in case of unbounded availability of tubulin.

Consistently with in vivo observations (Ehrhardt, 2008), we assume that microtubules are mainly—but not exclusively, created by microtubule-based nucleation. In particular, we divide the nucleation of new microtubules in two distinct nucleation types: microtubule-based nucleation and nucleation from a dispersed nucleation sites, with nucleation rates, respectively,

$$R_{n,b} = R_n\left(L_f\right) \frac{L_\parallel + L_\perp}{L_\parallel + L_\perp + L_*}, \tag{3}$$

and

$$R_{n,u} = R_n\left(L_f\right) \frac{L_*}{L_\parallel + L_\perp + L_*}. \tag{4}$$

$L_*$ is the *propensity length for dispersed nucleation*, that is, a constant that controls the fraction of nucleation events that occur from dispersed sites in the cytosol rather than from the lattice of already existing microtubules, and a proxy for the binding affinity of nucleation complexes to the microtubule lattice.

In the CA, microtubules that arise through microtubule-based nucleation are nucleated preferentially parallel or with an angle of about 40 with respect to the growth direction of the mother microtubule (Chan et al., 2009). This nucleation mechanism by itself can already contribute to maintaining the orientation of the CA (Deinum et al., 2011; Foteinopoulos & Mulder, 2014). As here, we only consider two possible directions for microtubules, we assume that new microtubules generated through microtubule-based nucleation have a strong bias towards growing in the same direction as the mother microtubule. We include this mechanism in our model by introducing the probability $z > 1/2$ for a new microtubule to be nucleated parallel to the mother microtubule and, consequently, $1 - z$ to be nucleated orthogonal to it. Then, the microtubule-based nucleation rates for new longitudinal and

transverse microtubules are, respectively,

$$R_{n,b}^{\parallel} = R_n\left(L_f\right) \frac{zL_\parallel + (1-z)L_\perp}{L_\parallel + L_\perp + L_*}, \tag{5}$$

and

$$R_{n,b}^{\perp} = R_n\left(L_f\right) \frac{(1-z)L_\parallel + zL_\perp}{L_\parallel + L_\perp + L_*}. \tag{6}$$

We reasonably assume that the medium is isotropic as regards the dispersed nucleation of new microtubules. Hence, the rates for dispersed nucleation are

$$R_{n,u}^{\parallel} = R_{n,u}^{\perp} = R_n\left(L_f\right) \frac{\frac{1}{2}L_*}{L_\parallel + L_\perp + L_*}. \tag{7}$$

It follows that the overall nucleation rates for the two populations are $R_n^{\parallel} = R_{n,u}^{\parallel} + R_{n,b}^{\parallel}$ and $R_n^{\perp} = R_{n,u}^{\perp} + R_{n,b}^{\perp}$ (see also Figure 1c).

**2.4. Dynamics of individual microtubules.** All microtubules are nucleated in the growing state, with growth speed $V^+$. Experimentally, the growth speed is approximately proportional to the amount of free tubulin (Wieczorek et al., 2015). Here, however, consistently with equation (2), we also allow for the inevitable saturation of the growth speed of microtubules, assuming it to be given by

$$V^+\left(L_f\right) = v^+ \frac{L_f}{L_f + L_v}. \tag{8}$$

We also note that in the limit of $L_v \gg L_f$, equation (8) implies a linear relation between growth speed and abundance of free tubulin, in which case our model recapitulates the experimental observations. Growing microtubules can switch from the growing to the shrinking state with constant catastrophe rate $r_c$. Microtubules in the shrinking state shrink with constant speed $v^-$, and they can switch from the shrinking to the growing state with rescue rate $r_r$.

When differently oriented microtubules cross each other, they create a *crossover*, where a severing event can take place. In particular, the occurrence or not of a severing event is partly influenced by the number of crossovers a microtubule has created and, consequently, to its length, and also partly by its relative position to the crossing microtubule. Indeed, experiments have shown that newer microtubules, that cross over the top of already existing ones, are preferentially severed (Lindeboom et al., 2013). Since our model consists of two populations of microtubules without a specific localisation in space, we introduce an effective severing rate for an individual microtubule to model the creation of crossovers and the occurrence of severing events, which is proportional to the product of its own length and the total lengths of microtubules in the direction perpendicular to it. In particular, such proportionality can be explained by the following argument: consider the cell wall to be the lateral surface of a cylinder with height $h$ and radius $r$, then a longitudinal microtubule of length $l$ crosses at least one transverse microtubule with probability $\frac{l}{h} \sum_{i=1}^{M_\perp} \frac{l_{\perp,i}}{2\pi r} = \frac{1}{2\pi rh} lL_\perp$. Furthermore, to be consistent with the experimental observations that new microtubules are, in general, longitudinally oriented, we introduce a preferential severing for the latter by assuming that $q > \frac{1}{2}$ is the fraction of longitudinal microtubules severed over the total number of severed microtubules. Then, if $r_s$ is the intrinsic severing rate at a crossover, the overall severing rates for longitudinal and transverse microtubules of length $l$ are, respectively,

$$R_{s,\parallel}(l) = q r_s l L_\perp, \tag{9}$$

and

$$R_{s,\perp}(l) = (1-q) r_s l L_\parallel. \tag{10}$$

Once a microtubule is severed, the newly created plus end of the lagging microtubule enters either the growing state with probability of stabilisation-after-severing $p^+$, or the shrinking state with probability $1 - p^+$, while the state of the plus end of the leading microtubule is unaffected. Experimental and theoretical work has shown that a relatively high probability of stabilisation-after-severing ($p^+ \simeq 0.15$) is required in order to commence the reorientation process (Lindeboom et al., 2019; Saltini & Mulder, 2020). Finally, the newly-created minus end of the leading microtubule remains stable, without undergoing dynamic instability, see Figure 1d.

Notice that without limitations in the abundance of free tubulin, a severing mechanism as described here would create an exponentially increasing number of both longitudinal and transverse microtubules (Saltini & Mulder, 2020), depriving our model of any biological significance.

### 2.5. Small increase in the growth speed of longitudinal microtubules

Previous experimental work on the effect of cell geometry on microtubule dynamics have revealed that the mechanical stress induced by cell wall geometry can influence the alignment of cortical microtubules (Hamant et al., 2008; Landrein & Hamant, 2013). Furthermore, several computational approaches have shown that slight directional cues caused by, for example, catastrophes or microtubule stabilisation induced by features of the surface geometry, can select a single preferential alignment of the CA (Chakrabortty et al., 2018; Mirabet et al., 2018).

Many plant cells, most notably those in the elongation zone of roots, have a roughly cylindrical morphology. This implies that they have a clearly distinguished direction of maximal wall curvature, transverse to the cell axis, and of minimal curvature, parallel to the cell axis. It is conceivable that this small but significant difference has an impact on microtubule dynamics. We take this observation as the starting point for our main hypothesis assuming that longitudinal microtubules grow slightly faster than transverse microtubules, and that such a bias is a sufficient ingredient to obtain a maintained reorientation of the CA. Therefore, we set

$$v_\parallel^+ = \alpha v_\perp^+ \equiv \alpha v^+, \tag{11}$$

with $\alpha > 1$ and $\alpha \simeq 1$. All other dynamic parameters of the model are unaffected. Our choice of applying a difference only in the growth speed is motivated by a practical reason. Indeed, changing only the growth speed makes the model also analytically tractable. Nevertheless, in Section 3, we will show that both increasing the intrinsic nucleation rate and decreasing the catastrophe rate in the longitudinal direction, which has an analogous effect on the relative stability of the transverse and longitudinal microtubules, leads to qualitatively similar results on the steady-state properties, suggesting that the choice of the specific dynamic parameter to vary in order to obtain a maintained reorientation is arbitrary.

### 2.6. Polarisation and transverse suppression

We refer to the initial number of transverse microtubules as $M_\perp^0$, and to the amount of tubulin incorporated in them as $L_\perp^0$. To evaluate the efficiency of the reorientation, we define the following order parameters:

- *number* and *length polarisation*, respectively,

$$\mathcal{P}_M = \frac{M_\parallel - M_\perp}{M_\parallel + M_\perp}, \qquad \mathcal{P}_L = \frac{L_\parallel - L_\perp}{L_\parallel + L_\perp},$$

as order parameters for the difference between the two populations and

- *number* and *length transverse suppression*, respectively,

$$\mathcal{R}_M = \frac{M_\perp^0 - M_\perp}{M_\perp^0 + M_\perp}, \qquad \mathcal{R}_L = \frac{L_\perp^0 - L_\perp}{L_\perp^0 + L_\perp},$$

as order parameters that estimate how much of the original transverse array is still present at the end of the process.

Ideally, to consider the reorientation as efficient, we have two requirements: (a) that all four-order parameters are close to 1, that is, the majority of microtubules and the amount of incorporated tubulin are polarised in the longitudinal direction and (b) that the time scale for the array to reorient is comparable to the experimentally observed one (Lindeboom et al., 2013, 2019).

## 3. Results

### 3.1. Computational approach

We run stochastic simulations of our model to assess whether the difference in growth speed between the two different populations is sufficient to achieve a full and maintained reorientation of the array. Simulations consist of an initial phase in which we obtain a steady-state transverse array, see Supplementary Information S2. Then, at time $t = 0$, we allow the nucleation of longitudinal microtubules

**Table 1.** Reference values for the parameters of the model

| Parameter | Description | Numerical value | Units |
|---|---|---|---|
| $v^+$ | Growth speed | 0.103 | μm/s |
| $v^-$ | Shrinkage speed | 0.225 | μm/s |
| $r_c$ | Catastrophe rate | 0.01 | $s^{-1}$ |
| $r_r$ | Rescue rate | 0.026 | $s^{-1}$ |
| $r_s$ | Severing rate | $2 \cdot 10^{-7}$ | $s^{-1} \, \mu m^{-2}$ |
| $r_n$ | Nucleation rate | 0.3 | $s^{-1}$ |
| $\alpha$ | Speed bias | 1.1 | – |
| $p^+$ | Stabilisation-after-severing | Tuned | – |
| $q$ | Fraction of severed longitudinal microtubules | Tuned | – |
| $L_{tot}$ | Total amount of tubulin | $4 \cdot 10^3$ | μm |
| $L_v$ | Crossover length | $8 \cdot 10^2$ | μm |
| $L_*$ | Propensity length for dispersed nucleation | $10^2$ | μm |
| $z$ | Probability of nucleation parallel to the parent | 0.9 | – |

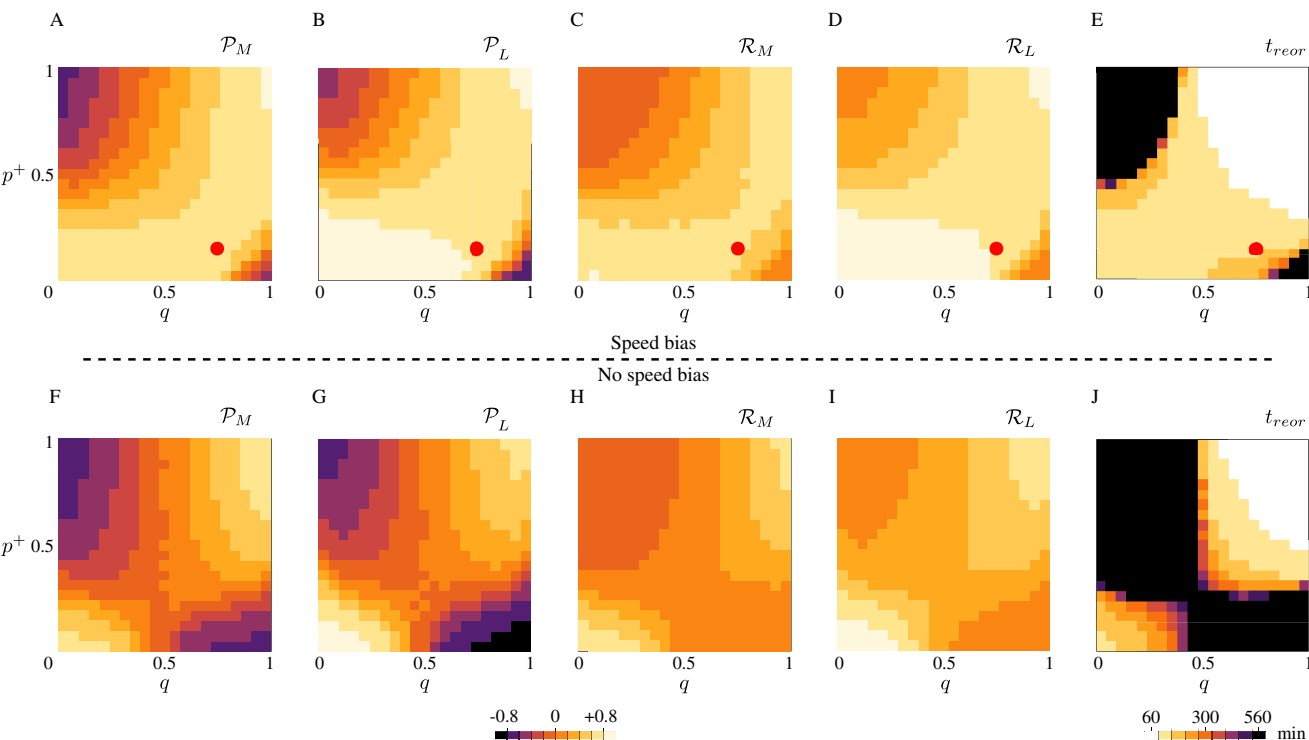

**Fig. 2.** (a, f) Microtubule number polarisation, (b, g) microtubule length polarisation, (c, h) transverse number suppression, (d, i) transverse length suppression and (e, j) transverse-to-longitudinal reorientation time as functions of $q$ and $p^+$. Lighter colors correspond to a more efficient reorientation. (a–e) Results for the case of bias in the growth speed of the longitudinal population over the transverse population. The red circles correspond to the parameters used to obtain Figure 3, that is, $q = 0.75$, $p^+ = 0.15$. (f–j) Results for the case of no bias in the speed of the two differently oriented populations. (a–d) The range of values for polarisation and suppression runs from −1 to +1. (e) Black areas in the $(q, p^+)$ plane correspond both to reorientation processes that required more than 540 min, or non-occurred reorientation. Results are averaged over $N = 10^3$ simulations.

both through microtubule-based nucleation and dispersed nucleation. We let the simulations run until the system reaches again a steady-state and record the number and length polarisation and suppression, and the time required to obtain a full reorientation of the CA. We perform a sensitivity analysis in the $(q, p^+)$ plane by separately tuning them from 0 to 1. We average the results over $N = 10^3$ stochastic simulations $per (q, p^+)$ couple. Parameters and relative numerical values used in the simulations are listed in Table 1. A brief description of the computer simulations can be found in Supplementary Information S3.

Numerical values for the dynamic parameters and the amount of tubulin has been chosen consistently with experimental measurements (Lindeboom et al., 2019; Wieczorek et al., 2015).

Figure 2a–d shows the polarisation, that is, the relative number of longitudinal microtubules (the relative amount of tubulin used by longitudinal microtubules, respectively) compared to the total, and the transverse suppression, that is, the relative number of transverse microtubules (the relative amount of tubulin used by transverse microtubules, respectively) compared to the initial array, for both microtubule number and length, whereas Figure 2e shows

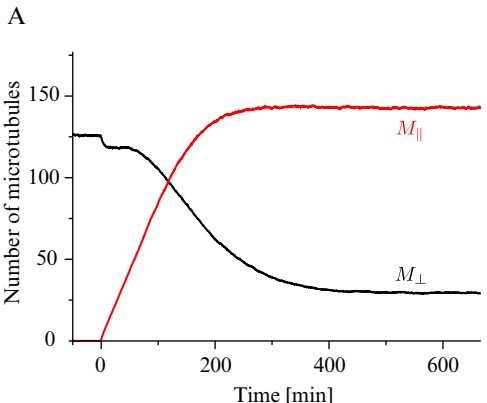
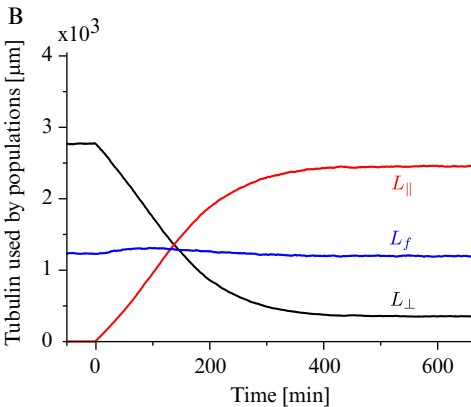

**Fig. 3.** Time evolution of (a) longitudinal (red) and transverse (black) microtubules and (b) tubulin used by the longitudinal population (red), the transverse population (black), and the free tubulin (blue), for $q = 0.75$, $p^+ = 0.15$. Results are averaged over $N = 10^3$ simulations.

the time required by the system to achieve the reorientation, in the case of bias in the growth speed of the longitudinal microtubules over the transverse microtubules. Figure 2f–j shows the same quantities measured in the case of identical growth speed for the two populations of microtubules. Lighter colors in the figure correspond to more efficient reorientation. Although the figure shows that a more efficient and fast reorientation requires high values of both the parameters governing preferential severing ($q$) and stabilisation-after-severing ($p^+$), we can still observe a good degree of reorientation for $p^+$ in the range from 0 to 0.25, that is, in the biologically relevant range of values for the probability of stabilisation-after-severing. In particular, the redistribution of tubulin seems to be very efficient in that range of values, see Figure 2b,d. It is also interesting to notice that in this regime, polarisation and suppression seem to be not strongly dependent on $q$. This suggests that, even though the preferential severing plays an important role in the amplification phase of the reorientation to boost the creation of longitudinal microtubules, in presence of a growth-speed bias it is not strictly required to maintain the new longitudinal array. From Figure 2e, it is interesting to notice that even in the case of very high values for both $q$ and $p^+$, the time needed by the system to achieve a full reorientation of the array is on the order of 1 hr, in apparent contrast with the experimental findings, where the reorientation time have been measured to be around 30 min. The reason of this apparent discrepancy, is that the proposed model, although it provides a good quantitative explanation of the steady-state properties of the system, is still too idealised to quantitatively reproduce the timescale of the process. Nevertheless, as on can infer by comparing Figure 2e with Figure 2j, the bias in the growth speed of longitudinal microtubules over transverse microtubules significantly reduces the timescale of the reorientation, thus qualitatively explaining that a bias in the growing speed of the two microtubule populations not only causes the longitudinal lock-in, but also influences the time needed by the system to reach the steady-state.

Figure 3 shows the time evolution of the number and the total length of microtubules belonging to the different populations. The combination of preferential severing and biased growth speed has a double effect: it makes the reorientation efficient by increasing the number of longitudinal microtubules at the steady-state and suppressing the transverse, and it boosts the speed at which the reorientation occurs.

We also tested the alternative hypothesis that the factors that promote the commencement of the reorientation process, namely,

high probability of stabilisation-after-severing and preferential severing for longitudinal microtubules (Lindeboom et al., 2019; Saltini & Mulder, 2020), could be sufficient to explain the experimentally observed maintained reorientation. Our simulations showed that, although the two ingredients are indeed necessary to quickly start the reorientation process and reach a steady-state, they cannot explain how the full reorientation is achieved and maintained (see Figure 2f–j and Supplementary Information S4).

### 3.2. Analytical approach

To further test our hypothesis that reorientation and maintenance of the array are caused by a biased recruitment of tubulin towards the longitudinal microtubules, we analytically study a simplified version of our model in which the two populations of microtubules compete for the tubulin pool without direct interaction between them, that is, without possibility of severing events. Therefore, we set $r_s = 0$. We make the further simplification of assuming complete depolymerisation of microtubules just after catastrophe events, that is,

$$\frac{\langle l \rangle}{v^-} \ll \frac{1}{r_r}, \qquad (12)$$

where $\langle l \rangle$ is the average length of microtubules. Finally, we assume that all microtubules are nucleated in the same direction of the mother microtubule, that is, $z = 1$. To ease the notation, we hereafter drop the direct dependency of $V^+$ and $R_n$ on $L_f$ whenever this is not strictly necessary. Under these assumptions we can rewrite the dynamic equations (S1–S4) from Supplementary Information S1 as

$$\frac{\partial}{\partial t} m_\parallel (t, l) = -\alpha V^+ \frac{\partial}{\partial l} m_\parallel (t, l) - r_c m_\parallel (t, l), \qquad (13)$$

$$\frac{\partial}{\partial t} m_\perp (t, l) = -V^+ \frac{\partial}{\partial l} m_\perp (t, l) - r_c m_\perp (t, l), \qquad (14)$$

with boundary conditions

$$\alpha V^+ m_\parallel (t, 0) = R_n^\parallel, \qquad (15)$$

$$V^+ m_\perp (t, 0) = R_n^\perp. \qquad (16)$$

We now consider the 0th and the 1st moment equations corresponding to equations (13) and (14), that is

$$\frac{d}{dt} M_\parallel (t) = R_n^\parallel - r_c M_\parallel (t), \qquad (17)$$

$$\frac{d}{dt}M_\perp(t) = R_n^\perp - r_c M_\perp(t), \tag{18}$$

$$\frac{d}{dt}L_\parallel(t) = \alpha V^+ M_\parallel(t) - r_c L_\parallel(t), \tag{19}$$

$$\frac{d}{dt}L_\perp(t) = V^+ M_\perp(t) - r_c L_\perp(t), \tag{20}$$

coupled with the conservation of total tubulin

$$\frac{d}{dt}L_f(t) = -\frac{d}{dt}\left[L_\parallel(t) + L_\perp(t)\right]. \tag{21}$$

Suppose that $\alpha = 1$. Then, because of the symmetry between longitudinal and transverse microtubules, when longitudinal microtubules start to be nucleated the overall nucleation rate of all microtubules, as well as their growth rates, remain the same as in the initial system with only the transverse array. Therefore, we can safely assume that, in that specific case, $L_f = \mathrm{const}$. Here, since $\alpha$ is slightly greater than 1, and making the reasonable assumption that $L_f$ is smooth in $\alpha$, it follows that

$$\frac{d}{dt}\left[L_\parallel(t) + L_\perp(t)\right] \simeq 0. \tag{22}$$

This last equation implies that all building material used by the newer longitudinal array comes from the already existing transverse one, in agreement with our computational findings, see Figure 3b.

In order to determine what controls the ultimate polarisation of the microtubule distribution, and hence the reorientation mechanism, we study the steady-state version of the moment equations (17–20). For the time-dependent solution of the model, see Supplementary Information S5. If we isolate $r_c M_\parallel$ and $r_c M_\perp$ from the four equations we obtain

$$R_n^\parallel = r_c M_\parallel = \frac{r_c^2}{\alpha V^+}L_\parallel, \tag{23}$$

and

$$R_n^\perp = r_c M_\perp = \frac{r_c^2}{V^+}L_\perp. \tag{24}$$

From equation (23), we can observe that multiplying $v^+$ by $\alpha$ for the longitudinal growing speed is equivalent to dividing $r_c$ by $\sqrt{\alpha}$. Furthermore, if we re-write equation (23) with the explicit expression for the nucleation rate of longitudinal microtubules, we obtain $r_n\frac{L_\parallel + L_*/2}{L_\parallel + L_\perp + L_*} = \frac{r_c^2}{\alpha V^+}L_\parallel$, from which we can notice that multiplying $v^+$ by $\alpha$ for the longitudinal growing speed is also equivalent to multiplying $r_n$ by $\alpha$.

If we define

$$\lambda_{\parallel/\perp} \equiv \frac{L_{\parallel/\perp}}{L_{tot} - L_f}, \tag{25}$$

$$\lambda_* \equiv \frac{L_*}{L_{tot} - L_f}, \tag{26}$$

and we divide equation (23) by (24), by making use of equations (5)–(7), we obtain the system

$$\begin{cases} \frac{\lambda_\perp}{\lambda_\parallel} = \frac{\frac{1}{2}\lambda_* + \lambda_\perp}{\alpha\left(\frac{1}{2}\lambda_* + \lambda_\parallel\right)}, \\ \lambda_\parallel + \lambda_\perp = 1. \end{cases} \tag{27}$$

The system can be solved to find

$$\lambda_\parallel = \frac{(\alpha-1) - \frac{1}{2}\lambda_*(\alpha+1) + \sqrt{\left[(\alpha-1) + \frac{1}{2}\lambda_*(\alpha+1)\right]^2 - 2\lambda_*(\alpha-1)}}{2(\alpha-1)}, \tag{28}$$

$$\lambda_\perp = \frac{(\alpha-1) + \frac{1}{2}\lambda_*(\alpha+1) - \sqrt{\left[(\alpha-1) + \frac{1}{2}\lambda_*(\alpha+1)\right]^2 - 2\lambda_*(\alpha-1)}}{2(\alpha-1)}. \tag{29}$$

If we divide both sides of equation (29) by $(\alpha-1) + \frac{1}{2}\lambda_*(\alpha+1)$ and we expand the square root we obtain

$$\lambda_\perp \simeq \frac{1}{2}\frac{\lambda_*}{(\alpha-1) + \frac{1}{2}\lambda_*(\alpha+1)}. \tag{30}$$

Consequently,

$$\lambda_\parallel \simeq 1 - \frac{1}{2}\frac{\lambda_*}{(\alpha-1) + \frac{1}{2}\lambda_*(\alpha+1)}. \tag{31}$$

By putting equations (30) and (31) into equations (23) and (24), we obtain the number of microtubules in the longitudinal and in the transverse directions

$$M_\parallel = \frac{R_n}{r_c}\frac{1}{\lambda_*+1}\left[\frac{1}{2}\lambda_*\left(1 - \frac{1}{(\alpha-1) + \frac{1}{2}\lambda_*(\alpha+1)}\right) + 1\right], \tag{32}$$

$$M_\perp = \frac{R_n}{r_c}\frac{\frac{1}{2}\lambda_*}{\lambda_*+1}\left(1 + \frac{1}{(\alpha-1) + \frac{1}{2}\lambda_*(\alpha+1)}\right). \tag{33}$$

Notice that, in the $\lambda_* \to 0$ limit—that is, when nucleation is only microtubule-based, all tubulin is polarised in the longitudinal direction, and we observe full reorientation of the array from the transverse to the longitudinal direction. Indeed $\lim_{\lambda_* \to 0}\lambda_\parallel = 1$, and $\lim_{\lambda_* \to 0}M_\parallel = \frac{R_n}{r_c}$. On the other hand, when $\lambda_* \to \infty$—that is, when nucleation is only dispersed,

$$\lim_{\lambda_* \to \infty}\lambda_\parallel = \frac{\alpha}{\alpha+1}, \tag{34}$$

$$\lim_{\lambda_* \to \infty}\lambda_\perp = \frac{1}{\alpha+1}, \tag{35}$$

$$\lim_{\lambda_* \to \infty}M_\parallel = \lim_{\lambda_* \to \infty}M_\perp = \frac{1}{2}\frac{R_n}{r_c}, \tag{36}$$

implying that, although in this limit, the isotropy of the dispersed nucleation imposes that the final number of microtubules of the two populations is the same, the $\alpha > 1$ bias in the growth speed still produces a slight tubulin polarisation in the longitudinal direction by virtue of the longitudinal microtubules being on average slightly longer than the transverse ones, see Figure 4 and Supplementary Information S6. The foregoing analysis shows that even in the absence of the severing process, the bias in the growth speed by itself is a sufficient driver of the shift towards the longitudinally polarised state.

## 4. Discussion

Our goal, here, was to propose a plausible mechanism by which the striking reorientation of the CA of dark-grown hypocotyl cells upon exposure to blue light can be "locked-in" on longer post-stimulus timescales. A major ingredient of the model we propose, the fact that tubulin availability is an important limiting factor,

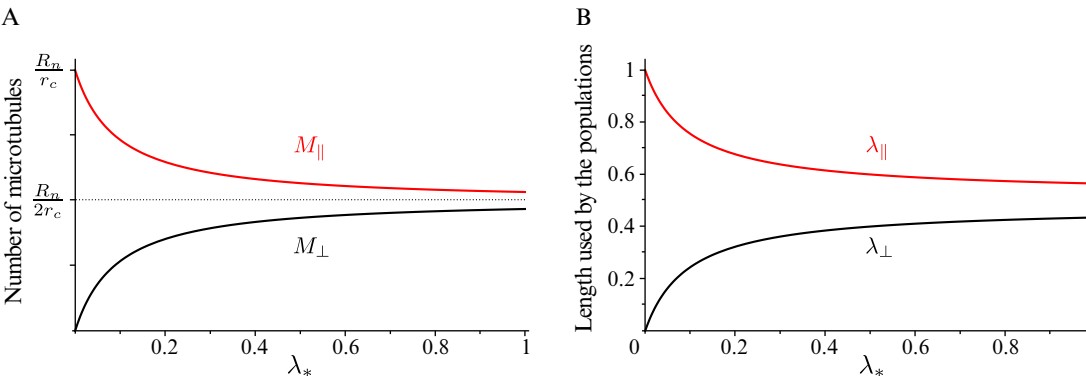

**Fig. 4.** (a) Number of microtubules and (b) non-dimensional length used by microtubules populations as functions of the propensity length for dispersed nucleation, with $\alpha = 1.1$.

follows naturally from the mechanism originally put forward to explain the onset reorientation, which involves the exponential amplification of longitudinal microtubules due to the initially predominant severing occurring at cross-overs with the pre-existing transverse array. This exponential increase will, at some time of necessity, deplete the pool of available tubulin. From that point on, the process essentially chokes itself and an inevitable redistribution of orientations towards a more isotropic equilibrium distribution follows. This points to the need for an additional assumption that explains the ultimate dominance of the longitudinal microtubules, which we provide in the form of the hypothesis that longitudinal microtubules are dynamically advantaged, be it through a slightly higher base growth speed, a slightly increased intrinsic nucleation rate, or a slightly decreased intrinsic catastrophe rate. This makes our proposed mechanism an interesting example of the generic competitive exclusion principle in population ecology (Hardin, 1960). According to that principle, two populations competing for the same limited resources cannot coexist at equilibrium. Instead, the competitively superior population will persist, while the other will go extinct. In our case, the strict validity of this principle is attenuated by three regulating factors that prevent the total extinction of the initial transverse array: the occasional occurrence of severing events for transverse microtubules, the few dispersed nucleation events in the transverse direction, and the microtubule-based nucleation of transverse from longitudinal microtubules. However, when none of those three factors is present, as in the $\lambda_* \to 0$ case of Section 3, only the longitudinal array would persist. In this light, our model does not explain the transverse-to-longitudinal reorientation in any geometrical sense, rather from a population dynamics perspective. The result that in the $\lambda_* \to 0$ limit it is possible to obtain a full reorientation of the CA even without the occurrence of severing events is somehow consistent with the results presented by Uyttewal et al. (2012), in which it was shown that, upon increasing the stress of microtubules through isoxaben treatment on meristematic cells, a certain degree of microtubule reorientation was achieved in mutants with decreased katanin activity. Interestingly, a computational study (Sambade et al., 2012) has predicted that the transition between different microtubule orientations could be caused by increasing the number of microtubules entering the outer cell face from the transverse sidewall, providing support to our proposed model.

It is somewhat striking that preferential severing for longitudinal microtubules and high probability of stabilisation-after-severing, two factors which have earlier been identified as crucial for the start of the reorientation process (Lindeboom et al., 2019; Saltini & Mulder, 2020), do not appear to be necessary for the long-term maintenance of the longitudinal array. However, the computational results presented here reveal that both preferential severing and high probability of stabilisation-after-severing play a key role in determining the speed of the reorientation.

Naturally, the model we propose raises some questions and issues, which we hope can be addressed by further research. Firstly, an experimental observation of a difference in any of the dynamic parameters between differently oriented microtubules is, at present, lacking. Nevertheless, such a difference could feasibly be revealed by a more detailed data analysis on microscopy movies of cortical microtubules of *A. thaliana* during the reorientation of the cortical array. The detection of an even small difference between the dynamics of longitudinal and transverse microtubules could potentially provide support for our theoretical predictions. Secondly, it begs for a mechanistic explanation, which we cannot at present provide. An obvious idea would be to appeal to a putative sensitivity of cortical microtubule dynamics to the local curvature or the mechanical state of the cell wall induced by this difference in curvature. Unfortunately, here, the evidence appears to point exactly into the opposite direction. Many observations suggest (Colin et al., 2020; Hamant et al., 2008; Uyttewaal et al., 2012) that microtubules consistently orient themselves along lines of maximal cell wall stress. In the type of cylindrical cell geometry, we envisage here, this stress is maximal in the *transverse* direction, forming the basis for the widely accepted explanation for the predominance of the transverse orientation of the cortical array. The molecular mechanism responsible for this preference is not as yet fully elucidated. However, these observations do imply that in principle microtubule stability can, potentially through molecular intermediaries, couple to the curvature of the cortex, leaving open the question whether this coupling is enhancing or depressing stability depending on the sign and magnitude of the curvature. In this light, it is intriguing to speculate whether the suggested difference in microtubule dynamics could in fact be caused by some downstream effect of the light exposure itself, which could activate an effector species that overrides the default coupling to the cell wall stress.

Secondly, the model as presented here is still highly idealised, and arguably needs to be developed further to include a more realistic description of the actual system. Specifically, our assumption that the transverse and longitudinal microtubule populations effectively live in distinct "spaces" and only interact at an ensemble level through the shared tubulin pool and the mutual modulation of the severing rates is a strong simplification. In reality, the common cell geometry which they share, the fact that they have continuous orientation, as well as the full repertoire of interactions at the level of individual microtubules such as, for example, induced

catastrophes due to collisions and zippering should be taken into account. Whilst the complexity of adding this level of detail will preclude an analytical approach, the computational tools for such a more comprehensive approach are available, for example, in the form of the framework described in Tindemans et al. (2010, 2014), and recently extended to deal with arbitrary cell geometries (Chakrabortty et al., 2018). Addition of a bias in the speed of growing microtubules depending on their orientation is readily implementable and, in principle, would allow for a more realistic computational test whether such a difference can indeed explain a reorientation of the array and its long-term maintenance.

## Acknowledgements

We thank Jelmer Lindeboom for a careful reading of the manuscript and for providing Figure 1a.

**Financial support.** The work of M.S. was supported by the ERC 2013 Synergy Grant MODELCELL. The work of B.M.M. is part of the research program of the Dutch Research Council (NWO).

**Conflict of interest.** The authors declare no competing interests.

**Authorship contribution.** M.S. designed and performed the simulations, and carried out the formal analysis. M.S. and B.M.M. conceived the study, designed the model, and wrote the manuscript.

**Data availability statement.** The simulations code that supports the findings of this study is available from the corresponding author, Marco Saltini, upon reasonable request.

**Supplementary Materials.** To view supplementary material for this article, please visit https://dx.doi.org/10.1017/qpb.2021.9.

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
