## [Reviewer Report]

Dear Editor,

We hereby submit our manuscript entitled: "A plausible mechanism for longitudinal lock-in of the plant cortical microtubule array after light-induced reorientation" for consideration at the journal Quantitative Plant Biology. 

The creation of new microtubules plays a crucial role in the reorganization of the cortical array of dark-grown hypocotyl cells of plant seedlings. Besides the nucleation of new polymers, their creation through severing upon exposure to blue light has been identified as the key mechanism to start the reorientation of the array from its standard direction transverse to the growth direction of the cell, to a longitudinal one. This mechanism is driven by katanin-mediated severing events at crossovers between differently oriented microtubules. Katanin preferentially severs longitudinal microtubules and, therefore, amplifies their number, resulting in the effective creation of a new, longitudinal array. However, all things being equal, one would expect that either the same mechanism occurs again and reorient the array from the new longitudinal direction to the original transverse one, or the two populations of microtubules reach an isotropic equilibrium when the availability of free tubulin naturally depletes as a consequence of its utilization by the newly-created microtubules. This is in contrast with in vivo observations where the reoriented longitudinal array is observed to be persistent over long time periods.

Our manuscript addresses the following question: what is the macroscopic mechanism behind the long term lock-in of the longitudinal array orientation?

To that aim, we make the concrete hypothesis that the persistence of the longitudinal array over long time periods is caused by a small orientation-dependent shift in microtubule dynamics. More specifically, we develop a stochastic model for two populations of dynamic microtubules (i.e., transverse population and longitudinal population) that compete for a finite pool of free tubulin. The shift in microtubule dynamics between the two populations is described by a small advantage for the longitudinal individual microtubules in terms of accessibility to the free tubulin pool. Our analytical calculations and computer simulations show that our proposed hypothesis indeed provides a sufficient mechanism to explain the long term lock-in of the longitudinal array orientation. Furthermore, in our manuscript, we show on the one hand that the alternative hypothesis that there is a selective advantage in severing longitudinal microtubules, is not strictly required to obtain a persistent longitudinal reorientation while, on the other hand it does contribute to the reorientation process by significantly accelerating it.

We believe that our work will be of interest to the journal Quantitative Plant Biology readership for a number of reasons, including:

1) The reorganization of the cortical microtubule array plays a central role in the growth-driven morphogenesis of the cell and, therefore, also of the plant as a whole. Our manuscript proposes a plausible mechanism that explains how the longitudinal lock-in of the array after light-induced reorientation can be maintained over a long time period, as observed in in vivo experiments.

2) Our assumption that a small orientation-dependent shift in microtubule dynamics exists can be feasibly be revealed by a detailed data analysis on the dynamics of cortical microtubules of A. thaliana during the light-induced reorientation of the array.

3) Our theoretical predictions have, in principle, the potential to motivate further research on the organization of the plant cellular cytoskeleton. Indeed, our findings could be confirmed by a more comprehensive computational approach that takes into consideration the full repertoire of interactions at the level of individual microtubules. Furthermore, should the orientation-dependent shift in microtubule dynamics be observed, such observation would call for further studies aimed at understanding the molecular mechanism responsible for the bias in the dynamics properties of microtubules as a consequence of their orientation.

4) Understanding the role of the dynamics of biomolecules is central to the study of many non-equilibrium stochastic processes that underlie biological function at the cellular scale. Therefore, we believe that our model will be of interest to the wider physics community interested in intra-cellular dynamical systems.

We are looking forward to your reply.

Yours sincerely,

Marco Saltini 

Bela Mulder

---

## [Reviewer Report]

*Comments to Author*: This paper nicely shows how a switch to longitudinal microtubule alignment can be maintained through longitudinal microtubules growing slightly faster than transverse microtubules. Competition between alternative alignments for a limited tubulin pool then leads to the longitudinal array outstripping the transverse. The paper raises important issues concerning potential mechanisms for stabilising reorientations and is well executed. It would greatly help, however, if the biological implications of the paper were clarified.

1. To illustrate the importance of the limited tubulin pool, the authors might describe what would happen without such a limitation.

2. Figure 2 could be better explained in the Results text. A brief summary of what each measure means and how it is calculated would be helpful. Although this information is covered elsewhere it would help the reader follow the logic more readily. I suggest the position of q=0.75, p+ = 0.15 later used in Fig. 3 should be indicated with a cross or equivalent in Fig.2 to help with cross reference.

3. Consider moving Fig. 5 into Fig. 2 to allow comparisons and discuss in text.

4. Bottom of page 8 becaulabelanalytical typo.

5. The discussion for the biological basis of the longitudinal growth advantage could be clarified. Are the authors proposing this reflects the flextural rigidity of microtubules as others have proposed, or a local wall curvature-sensing mechanism? The discussion of stresses is confusing because these would confer a transverse rather than longitudinal advantage, or are authors suggesting that sometimes high wall may confer a growth disadvantage? A clearer explanation of how the authors envisage how geometry may influence microtubule growth rates would be helpful.

---

## [Reviewer Report]

*Comments to Author*: This manuscript investigates possible small alterations to microtubule dynamics that may explain observed switches in their organisational patterns, i.e. transversal vs longitudinal array orientations. Each orientation appears to be stable (locked-in) and each configuration is built from the same core components, so how this combination of dynamics with two stable points arises is not clear. The manuscript addresses this interesting and important problem.

The manuscript is well written. The problem is introduced in an accessible manner and the methodology is clear. This contribution builds on a series of important advances from the same group over many years. The model is well-motivated and is an elegant way to reducing a complex problem to something simpler and tractable, yet still captures the essence of the problem.

I have a few comments and suggestions for the authors to consider.

1) As I understand it, the actual geometry of the system does not enter into the model. This is a strength of the approach and how the authors have abstracted their model. This being the case, the characterisation of "transverse" and "longitudinal" are merely labels and could equally well be “1” and “2”, “A” and “B”, etc. If this is so, then their model shows how two competing populations compete for the same resource (as they nicely refer to in their Discussion) but has little to do with anything actually being transverse or longitudinal. I think this point could be made clearer, i.e. their model is a great example of how populations can compete and change dominance but it would be important to point out that is doesn't explain a transverse to longitudinal orientation switch in any geometric sense.

2) After equation (8) it is written that Lv << 1. I think more relevant would be Lv << Lf from the equation (8).

3) Stating that plant cells are cylindrical seems quite a generalisation and might benefit from for which tissues/cells this is a reasonable approximation.

4) The authors hypothesise a minor change in MT dynamics through a velocity speed adjustment. This suggestion isn’t a requirement but it would be nice to put this in context of Sambade et al (2012) TPC. Within experimental limits, Sambade et al failed to detect any differences in speeds between transverse and longitudinal MT and built a model with a bias in number of MT from the transverse side walls that explained their observations. Given the experimental limitations in detecting such small differences as proposed here, this data is very much compatible with the current model. However, it would be interesting to see whether the current model also works for a bias in nucleation rate of one population over the other?

5) On page 7, the authors suggest that a growth speed bias is not strictly necessary to maintain the new longitudinal array. This seems plausible and looks evident from Figure 2, however, small changes are claimed to be important overall so it would be good to validate this hypothesis but running the model strictly without a growth speed bias.

6) On page 7, the authors claim that the building material for the longitudinal arrays comes from the transverse arrays but as I understood the model I was not convinced by this. Rather, their observation seems to be a consequence of their approximate step-function at Lf in equation (2) which leads to Lf being approximately constant at Lv. Transverse arrays thus need to depolymerise in order for Lf to increase. When there is sufficient Lf then there is source material for the longitudinal array but it seems to me that the tubulin elements used in the longitudinal array needn’t have come from the transverse array per se.

7) Before equation (13) there is some extra punctuation. After equation (21) there seems to be copy and paste error.

8) There are some relevant papers that could be mentioned and referenced, eg. Allard (2010) Mol Bio Cell, Sambade et al (2012) TPC, Ambrose et al (2011) Nat Comm, …

9) A brief description of the MC calculations would be helpful.

10) Fig. 1a indicates that the reorientation occurs on a timescale > 1hr, but Fig. 2E, 3A, 5E all have reorientation occurring < 1 hr. Even in the extreme case where q=1, the timescale is not achieved. Can the authors comment on this and what it means for their mechanism or parameters?

11) Some further explanation of equations 9 and 10 would be helpful. I think it would be preferable to not mix rates and probabilities in the equations.

12) Perhaps the authors can speculate in the Discussion about the mechanisms for how light might induce the suggested changes.

---

## [Reviewer Report]

*Comments to Author*: Dear Prof Mulder

Thank-you for your submission to QPB. Your manuscript has been read by 2 reviewers and their comments are enclosed.

We welcome a revised version of this manuscript addressing the issues raised.

Best wishes

George Bassel

---

## [Reviewer Report]

*Comments to Author*: The authors have done a very thorough and excellent revision in which all my previous comments and questions have been addressed. I congratulate the authors on their work and their manuscript.

---

## [Reviewer Report]

*Comments to Author*: Dear Marco and Bela

Thank-you for submitting your revised manuscript and addressing the issues raised by both reviewers. We are pleased to accept this manuscript for publication in QPB.

Best wishes

George